# Factors associated with willingness to take COVID-19 vaccine among pregnant women at Gondar town, Northwest Ethiopia: A multicenter institution-based cross-sectional study

Zewdu Bishaw Aynalem[1]*, Tewodros Worku Bogale[2], Getasew Mulat Bantie[3], Agumas Fentahun Ayalew[4], Workineh Tamir[5], Dejen Getaneh Feleke[6], Birhaneslasie Gebeyehu Yazew[1]

1 Department of Nursing, College of Medicine and Health Sciences, Injibara University, Injibara, Ethiopia,
2 Department of Midwifery, College of Medicine and Health Sciences, Injibara University, Injibara, Ethiopia,
3 Community Health Faculty, Alkan Health Science, Business and Technology College, Bahir Dar, Ethiopia,
4 Department of Epidemiology, College of Medicine and Health Sciences, Injibara University, Injibara, Ethiopia, 5 Department of Medical Laboratory Science, College of Medicine and Health Sciences, Injibara University, Injibara, Ethiopia, 6 Department of Pediatrics and Child Health Nursing, College of Health Sciences, Debre Tabor University, Debre Tabor, Ethiopia

* zest7@yahoo.com

## Abstract

### Background

Coronavirus disease has spread worldwide since late 2019. Vaccination is critical in controlling this pandemic. However, vaccine acceptance among pregnant women is not well-studied. Therefore, this study aimed to assess the COVID-19 vaccine acceptance and associated factors among pregnant women attending antenatal care clinics in Gondar town, Northwest Ethiopia.

### Methods

An institution-based cross-sectional study was conducted among pregnant women attending antenatal care clinics at Gondar town, Northwest Ethiopia, 2021. About 510 study subjects were selected using a systematic random sampling technique from August 25 to September 10/2021. Data collection was done by using an interviewer-administered, structured questionnaire. Epi-info 7.2 was used to enter data and then exported to SPSS version 25 software for analysis. Bivariable and multivariable binary logistic regression models were used to identify factors associated with the outcome variable. Variables with a p-value < 0.2 in the bivariable analysis were entered into the multivariable analysis to control for possible confounders. Statistical significance is determined using an adjusted odds ratio and 95% confidence interval (CI) at a p-value of < 0.05.

**Data Availability Statement:** All relevant data are within the manuscript.

**Funding:** The author(s) received no specific funding for this work.

**Competing interests:** The authors have declared that no competing interests exist.

**Abbreviations:** ACOG, American College of Obstetricians and Gynecologists; ANC, Antenatal Care; CDC, Centers for Disease Control and Prevention; CMHS, College of Medicine and Health Sciences; COVID-19, Corona Virus Disease; CSA, Central Statistics Agency; ICU, Intensive Care Unit; SD, Standard Deviation; SPSS, Statistical Package for the Social Science; US, United States; WHO, World Health Organization.

## Results

Of 510 participants, 211 (41.4%) were willing to take COVID-19 vaccines. Maternal age $\geq$ 35 years (AOR: 5.678, 95% CI: 1.775–18.166), having contact history with COVID-19 diagnosed people (AOR: 7.724, 95% CI: 2.183, 27.329), having a pre-existing chronic disease (AOR: 3.131, 95% CI: 1.700–5.766), good knowledge about COVID-19 vaccine (AOR: 2.391, 95% CI: 1.144, 4.998) and good attitude towards COVID-19 vaccine (AOR: 2.128, 95% CI: 1.348) were significantly associated with the outcome variable.

## Conclusions

The willingness to take COVID-19 vaccine among pregnant mothers was low. Age, contact history with COVID-19 diagnosed people, chronic disease, knowledge, and attitude towards COVID-19 vaccine were factors associated with COVID-19 vaccine willingness. To enhance the COVID-19 vaccine acceptance, the government with different stakeholders should strengthen public education about the importance of getting COVID-19 vaccine.

## Introduction

Coronavirus disease (COVID-19) is a highly contagious virus spreading rapidly throughout the world since late 2019. As of December 11, 2021, there were over 269.5 million confirmed cases of COVID-19 worldwide, with over 5.3 million deaths [1]. Every day, new COVID-19 infections are reported in Ethiopia, making it one of the five African countries with the highest COVID-19 burden [2, 3]. As of December 11, 2021, the COVID-19 pandemic has 372,868 confirmed cases and 6,922 deaths in Ethiopia [1, 2].

Globally, more than 100 million women are pregnant at the moment [4]. For these women, COVID-19 is a threat to themselves and their babies [5, 6]. Pregnant women are more at risk of COVID-19 infection, with recent evidence showing that they are up to 70% more susceptible. Once infected, they are at a higher risk of developing complications such as intensive care unit (ICU) admission, mechanical ventilation, and death, as well as an increased risk of cesarean section, and post-partum depression [7, 8]. Moreover, data showed that pregnant women infected with COVID-19 are at a higher risk for preterm birth and pregnancy loss, ranging from 10% to 25%, and even up to 60% in women with a critical illness [9].

To prevent the abovementioned outcomes, the American College of Obstetricians and Gynecologists (ACOG) and the Centers for Disease Control and Prevention (CDC) highly recommend all pregnant and lactating women get the COVID-19 vaccine [10, 11] because this vaccine can prevent severe illness, death, and pregnancy complications related to COVID-19 infection [12]. Similarly, the World Health Organization (WHO) has approved more than three COVID-19 vaccines to date to combat the disease's spread and possible threat [13]. In addition, the Ethiopian government and Ministry of Health have worked hard in disseminating information on COVID-19 prevention measures [14] and have received 8.1 million COVID-19 vaccines from the COVAX facility [15]. Despite these efforts made, vaccine hesitancy and unwillingness to accept the COVID-19 vaccine have emerged as one of the world's most pressing issues [16, 17], particularly in pregnancy [18].

The refusal to accept the COVID-19 vaccine is estimated to be high and greatly differs between countries [19]. For example, an online survey of pregnant women in 16 countries found that COVID-19 vaccine acceptance was more than 80% in Mexico and India, but lower

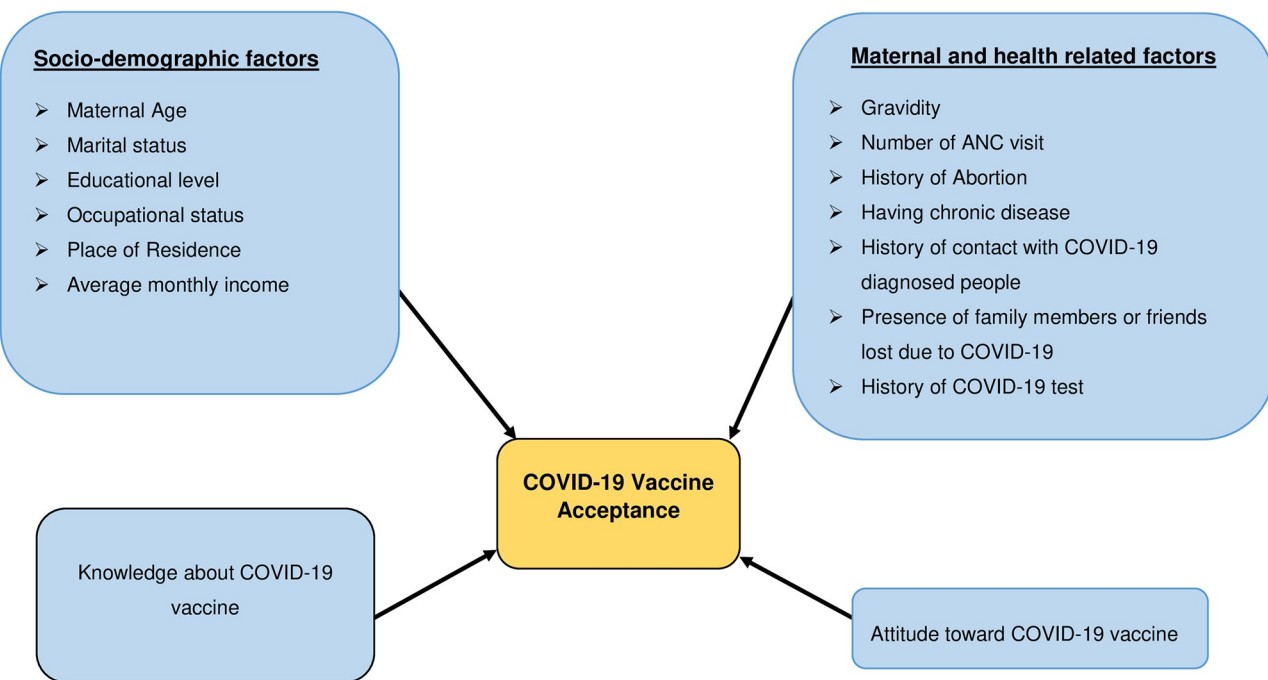

**Fig 1. A conceptual framework for COVID-19 vaccine acceptance and associated factors among pregnant women at Gondar town, Northwest Ethiopia, 2021.**

than 45% in the United States, Australia, and Russia [20]. In Switzerland [7], the United Kingdom [21], China [22], and Ethiopia [23], the willingness to accept COVID-19 vaccine among pregnant women was found to be 62.1%, 29.7%, 77.4%, and 70.7% respectively [7, 21, 22]. Given that the global intention for COVID-19 vaccination among pregnant women was estimated to be 47%, Africa had the lowest (19%) acceptance rate [24]. The main factors that determine an individual's intent to take the COVID-19 vaccine include being pregnant, gender, age, educational status, occupational status, income, pre-existence of chronic disease, attitude towards and knowledge of COVID-19 [23, 25–30] (Fig 1). Moreover, multiple myths and conspiracy theories about vaccines and COVID-19 [5, 31, 32] would also potentially affect the COVID-19 vaccine acceptance.

In Ethiopia, regardless of COVID-19 infection increment, people's preventive measures are insufficient to combat the pandemic [33, 34]. In this case, COVID-19 vaccination is critical for reducing pandemic transmission [35]. Though Ethiopia aims to vaccinate 20% of the population by the end of 2021 [36] still only 1.43% of them were fully vaccinated [15]. Furthermore, the intention of pregnant women to accept COVID-19 vaccination and the factors affecting it are not well-studied in this study area. As a result, assessing a pregnant woman's intent to receive the COVID-19 vaccine along with the main factors could help planners to increase awareness and assure people of the safety and benefits of vaccines, which, in turn, would help to control the spread of the virus and alleviate the negative effects of this pandemic [37, 38]. As well, as global vaccination rollouts continue, officials will also be able to utilize data on acceptance and predictors to better understand public intent and make informed policies [20]. Additionally, to the best of our knowledge, this is the first study in Northwest Ethiopia investigating COVID-19 vaccine acceptance among pregnant women. Therefore, to close this gap the current study was aimed to assess the COVID-19 vaccine acceptance and associated factors among pregnant women attending antenatal care (ANC) clinics in Gondar town, Northwest Ethiopia.

## Material and methods

### Study design, setting, and period

An institutional-based cross-sectional study was conducted among pregnant women attending antenatal care clinics in Gondar town, from August 25 to September 10, 2021. The town is located about 727 km from Addis Ababa, the capital of Ethiopia, and 180 km from Bahir Dar, the capital of Amhara State. In 2017, according to the Central Statistics Agency (CSA) population projection, the population of Gondar was estimated to be 360,600 [39]. There are 9 public health institutions in the town.

### Population, sample size determination, and sampling procedure

The source population for this study consisted of all pregnant women who live in Gondar town, Northwest Ethiopia. The study population for this study was pregnant women who attended antenatal care at selected public health institutions. Private health institutions were excluded from the study to avoid double counting since pregnant women could attend antenatal care in private institutions. Participants who were absent or critically ill during the data collection period were also excluded. The sample size was determined using a single population proportion formula by considering a 95% confidence level, a 5% margin of error, and a 70.7% COVID-19 vaccination acceptance [23]. The final sample size was 525 after accounting for the design effect of 1.5 and a 10% non-response rate. Of a total of nine public healthcare facilities, three of them were selected at random. Samples were proportionally assigned to each of the three selected health facilities. The number of pregnant women who attended antenatal care at selected public health facilities on average was 956 per month. To select each study participant, first, the sampling interval was calculated as follows: $K = N/n$, $K = 956/525 = 1.82 \approx 2$. Then, the first participant, participant 2, was selected randomly. Finally, based on the order of every $2^{nd}$ interval (i.e., $2^{nd}$, $4^{th}$, $6^{th}$, $8^{th}$ . . .) a participant was selected using the antenatal registration book via a systematic random sampling technique (Fig 2).

### Data collection procedure

Six Midwives with Bachelor's degrees were recruited to collect data (four for data collection and two for supervision). A structured and pre-tested questionnaire was used to collect data in a face-to-face interview. The interview took place after the woman received the ANC service, and each woman was interviewed privately and assured of confidentiality. During data collection, data collectors and supervisors were instructed to use face masks, maintain physical distance, and use hand sanitizer per WHO COVID-19 preventive measures. Every day, the investigators collected the completed questionnaires and checked them for completeness and consistency.

### Data collection instrument and study variables

The data collection tool was prepared by the investigators after an exhaustive literature search [23, 25, 40–43]. The questionnaire was prepared in English and translated to the local Amharic language and then back to English by language professionals. The questionnaire contains socio-demographic characteristics, knowledge about and attitude toward the COVID-19 vaccine, and intention of COVID-19 vaccine acceptance among pregnant women. The reliability of the tool was checked using Cronbach's alpha coefficient, the values were 0.87 for the overall tool, 0.73 for attitude subscale, and 0.82 for knowledge subscale and its content validity was assessed by public health specialists.

**COVID-19 vaccine knowledge.** To measure vaccine knowledge, 10 knowledge questions were asked. Each question had 2 possible answers: 1 for "no" and 2 for "yes." Participants who

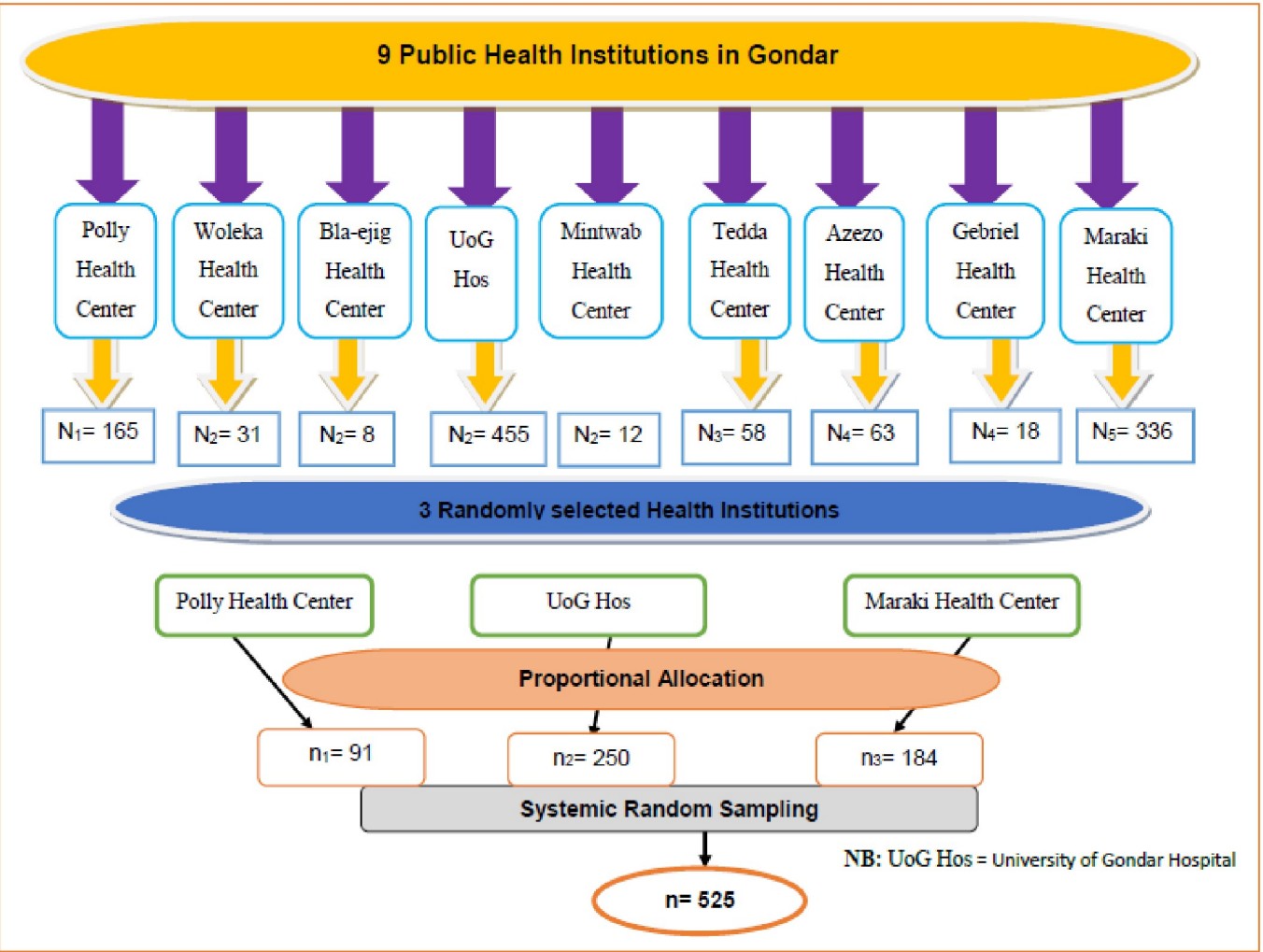

**Fig 2. Schematic presentation of sampling procedure about COVID-19 vaccine acceptance and associated factors among pregnant women at Gondar town, Northwest Ethiopia, 2021.**

correctly answered 80% or more of the knowledge questions were considered as having "good knowledge" otherwise "poor knowledge" [23, 42].

**Attitude about COVID-19 vaccine.** Study participant's vaccine attitude was also measured using 10 attitude questions. Each question had a 5-point Likert scale (1 was assigned for strongly disagree, 2 for disagree, 3 for neutral, 4 for agree, and 5 for strongly agree). Subsequently, participants who responded to ≥80% of the attitude-related questions were categorized as having "favorable attitude" if not as "unfavorable attitude" [23, 42].

**COVID-19 vaccine acceptance.** It represents the main outcome of the study and participants were asked, "Do you have an intention to accept the COVID-19 vaccine?" Those pregnant women who answered "Yes" to this question were considered vaccination accepters whereas those who answered "No" were considered vaccine hesitancy [23, 43].

## Quality control

To ensure data quality, the primary investigators provided two days of training to data collectors and supervisors about the study objectives, how to conduct interviews, and how to fill out

the questionnaire. The questionnaire was pretested on 53 study participants (10% of the sample size) with similar demographic characteristics. Accordingly, some language and typographic corrections have been addressed as needed. Supervision was made on daily basis by supervisors and principal investigators.

## Data processing and analysis

Epi-info 7.2 was used to enter data, which was then exported to SPSS version 25 software. Descriptive statistics including the means, standard deviations, frequencies, and percentages, were calculated and the finding of the analysis was presented in text, tables, and graphs. A binary logistic regression analysis was used to identify the association between dependent and independent variables. Model fitness was checked using a Hosmer-Lemeshow goodness-of-fitness test (p-value = 0.352). Again, to check the correlation between independent variables, multicollinearity test was carried out using variance inflation factor (VIF) and found 2.56. This VIF value confirmed the absence of significant collinearity among independent variables. Variables having a p-value of $\leq 0.2$ during bi-variable regression analysis were entered into multivariable regression analysis. Variables having a p-value $\leq$ of 0.05 in the multivariable analysis were taken as significant predictors of the outcome.

## Ethics approval

Ethical clearance was obtained from the ethical review committee of University of Gondar, College of Medicine and Health Sciences (CMHS) ethical review committee (S/N/1.6/10/2013). Before the beginning of data collection, a permission letter was obtained from the health department of Gondar town and each health facility. After the purpose and objective of the study had been explained, written consent was obtained from each study participant. Names of participants and other personal identifiers were not included in the data collection tool.

## Results

### Socio-demographic characteristics of the study participants

A total of 510 pregnant women were interviewed, with a response rate of 97.1 percent. The mean age of study participants was 30.7 (SD ±5.86) years, of whom 72.4% were between 25 and 34 years. Regarding the place of residence and educational status, 395 (77.5%) of the participants were city dwellers and 242 (47.5%) had secondary and above educational levels. The majority of the study participants, 284 (55.7%), were employed (Table 1).

### Maternal health care service characteristics of participants

Nearly two-thirds (329; 64.5%) of pregnant mothers were multigravida and 71 (13.9%) had at least one abortion history. All participants had an antenatal care follow-up. Of these, only 99 (19.4%) of mothers had 4 times ANC follow-up. Regarding medical illness, 74 (14.5%) mothers did have any history of medical illness (Table 2).

### Study participants' experience with COVID-19

Of 510 study participants, the majority of them (465; 91.2%) had no history of contact with COVID-19 diagnosed people. Four hundred sixty-nine (92%) of them said no one in their household had been lost to COVID-19. Of 166 tested participants for COVID-19, 12 (7.2%) had positive test results (Table 3).

**Table 1. Socio-demographic characteristics of pregnant woman attending antenatal care clinic at Gondar town, Northwest Ethiopia, 2021 (n = 510).**

| Variables | Category | Frequency | Percentage |
|---|---|---|---|
| Maternal age in years | 18–24 | 110 | 21.6 |
| | 25–34 | 369 | 72.4 |
| | ≥35 | 31 | 6.1 |
| Place of residence | Urban | 395 | 77.5 |
| | Rural | 115 | 22.5 |
| Marital status | Married | 437 | 85.7 |
| | Unmarried | 51 | 10 |
| | Others[M] | 22 | 4.3 |
| Educational status | No formal education | 109 | 21.4 |
| | Primary education | 159 | 31.2 |
| | Secondary and above | 242 | 47.5 |
| Occupational status | Employed | 284 | 55.7 |
| | Unemployed | 226 | 44.3 |

**Note:** *Others[M]* *includes (Divorced, Widowed, and Separated).*

**Table 2. Obstetric health care service characteristics of pregnant women attending antenatal care clinic at Gondar town, Northwest Ethiopia, 2021 (n = 510).**

| Variables | Categories | Frequency | Percentage |
|---|---|---|---|
| Gravidity | Primi-gravida | 181 | 35.5 |
| | Multigravida | 329 | 64.5 |
| Number of ANC visit | 1 time | 48 | 9.4 |
| | 2 times | 174 | 34.1 |
| | 3 times | 189 | 37.1 |
| | 4 times and above | 99 | 19.4 |
| History of Abortion | Yes | 71 | 13.9 |
| | No | 439 | 86.1 |
| Having known chronic illness | Yes | 74 | 14.5 |
| | No | 436 | 85.5 |
| Type of medical illness (n = 74) | Hypertension | 22 | 29.7 |
| | Diabetes | 20 | 27.0 |
| | Renal problem | 17 | 23.0 |
| | Others[MI] | 15 | 20.3 |

**Note:** *Others[MI]* *includes (HIV/AIDS and Cardiac Diseases).*

**Table 3. Participants' experience with COVID-19 among pregnant woman attending antenatal care clinic at Gondar town, Northwest Ethiopia, 2021 (n = 510).**

| Variables | Category | Frequency | Percentage |
|---|---|---|---|
| Having history of contact with COVID-19 diagnosed people | Yes | 45 | 8.8 |
| | No | 465 | 91.2 |
| The presence of family members or friends lost to COVID-19 | Yes | 41 | 8 |
| | No | 469 | 92 |
| Have tested for COVID-19 | Yes | 166 | 32.5 |
| | No | 344 | 67.5 |
| Result of COVID19 test (n = 166) | Positive | 12 | 7.2 |
| | Negative | 154 | 92.8 |

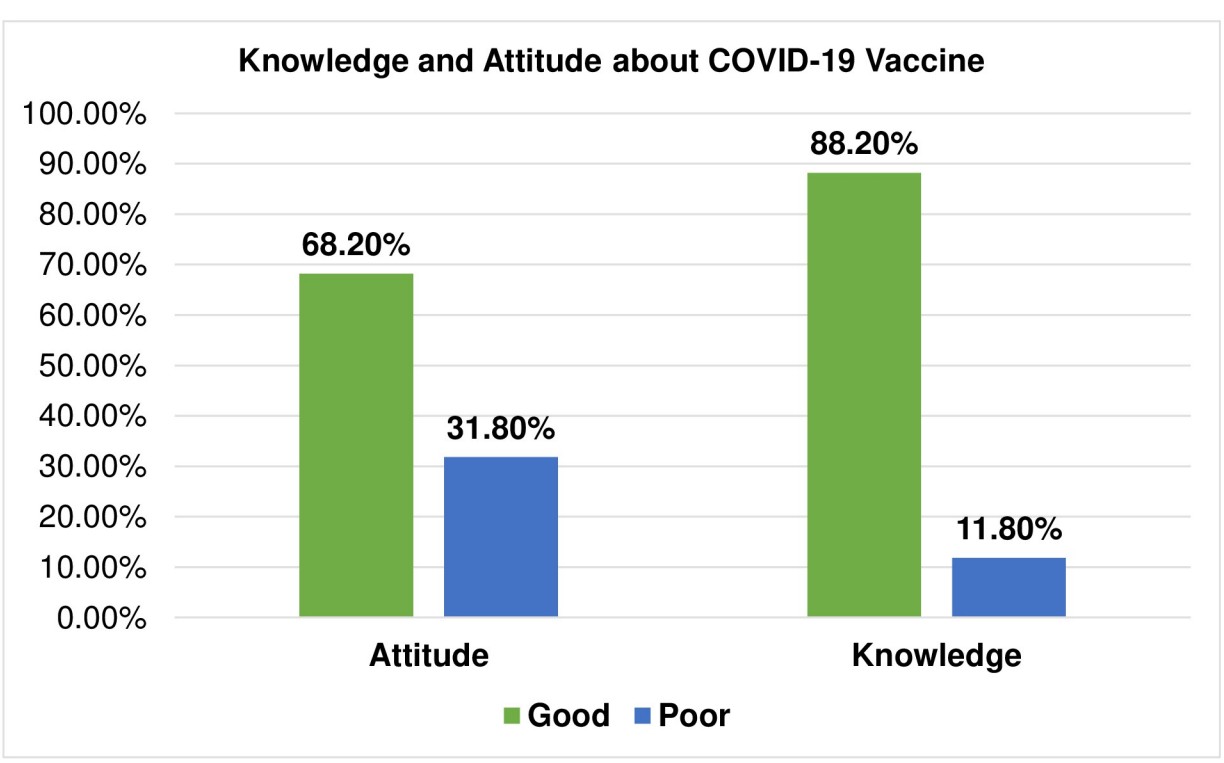

**Fig 3. Participants' knowledge and attitude about COVID-19 vaccine among pregnant woman attending antenatal care clinic at Gondar town, Northwest Ethiopia, 2021 (n = 510).**

### Knowledge and attitude towards COVID-19 among study participants

In this study, 450 (88.2%) and 348 (68.2%) mothers had good knowledge and a good attitude about COVID-19 vaccines respectively (Fig 3).

### COVID-19 vaccine acceptance among study participants

In this study, the COVID-19 vaccine acceptance was found to be 41.4% (95% CI; 37.1%–45.5%) among study participants. While 299 (58.6%) of them did not accept to use the COVID-19 vaccine. One hundred seventy-nine (35.1%) were concerned about the vaccine causing them or their unborn baby harm (Fig 4).

### Factors associated with COVID-19 vaccine acceptance

Of nine candidate variables entered into multivariable analysis, five of them namely: maternal age, knowledge about covid-19 vaccine, attitude towards a COVID-19 vaccine, having a pre-existing medical illness, and having contact history with COVID-19 diagnosed patient were found to be significantly associated with COVID-19 vaccine acceptance among pregnant women attending antenatal care clinics. The odds of accepting COVID-19 vaccine among pregnant women of ≥35 years age group were nearly 5.7 times more likely than those pregnant women found in the age group between 18–24 years (AOR: 5.678, 95% CI: 1.775–18.166).

Pregnant women who had good knowledge about COVID-19 vaccine were almost 2.4 times more likely to accept the COVID-19 vaccine compared to those women who had poor knowledge (AOR: 2.391, 95% CI: 1.144, 4.998). Compared with those pregnant women who had a poor attitude towards COVID-19 vaccine, women who had a good attitude toward

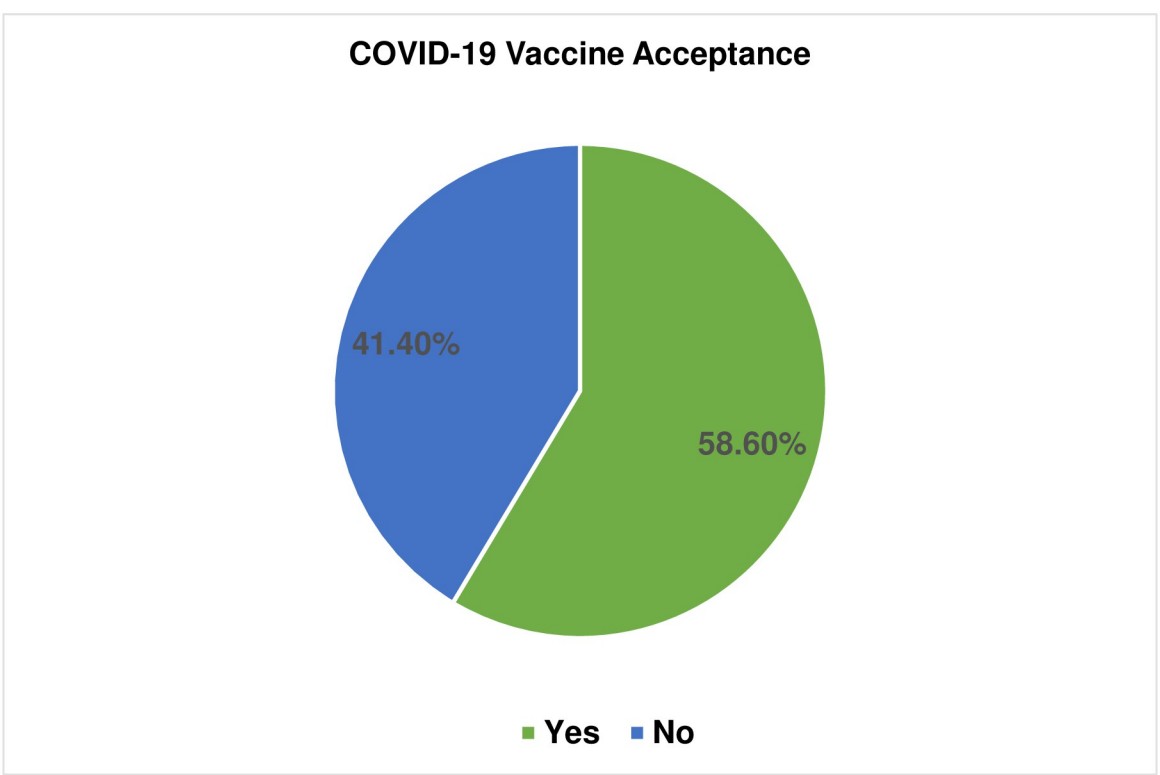

**Fig 4. COVID-19 vaccine acceptance among pregnant woman attending antenatal care clinic at Gondar town, Northwest Ethiopia, 2021 (n = 510).**

COVID-19 vaccine were 2.1 times (AOR: 2.128, 95% CI: 1.348, 3.360) more likely to have an intent for COVID-19 vaccination.

Those pregnant women who had a medical illness or pre-existing chronic disease were 3.1 times more likely to accept the COVID-19 vaccine compared to pregnant women who had no such problem (AOR: 3.131, 95% CI: 1.700–5.766). Moreover, pregnant women who had a contact history with COVID-19 diagnosed patients were roughly 8 times (AOR: 7.724, 95% CI: 2.183, 27.329) more likely to have an intention for COVID-19 vaccination than those who had no (Table 4).

## Discussion

COVID-19 vaccines have been launched as the ideal solution for bringing the pandemic of coronavirus diseases to an end in the world [44, 45]. For better implementation of the program, assessing the COVID-19 vaccine acceptance have a decisive role [46, 47]. Therefore, this study was aimed at assessing the COVID-19 vaccine acceptance and associated factors among Pregnant Woman Attending Antenatal Care Clinic at Gondar town, Northwest Ethiopia.

In this study, the acceptance of COVID-19 vaccine among pregnant women was only 41.4% (95% CI: 36.9–45.7). This finding was lower than studies conducted in 16 countries across the world (52.0%), Southwest Ethiopia (70.7%), global acceptance rate (47%), New York (58.3%), and Italy (52.9%) among pregnant women [20, 23, 24, 26, 48]. Correspondingly, in this study, COVID-19 vaccine acceptance among pregnant women was lower than the global acceptance range (55–90%) [49] and other studies conducted in Ethiopia (80.9% and 62.6%), China (91.3%), South Africa (81.6%), African countries (48.93%), and Indonesia (93.3%)

**Table 4. Bivariable and multivariable analysis of factors associated with COVID-19 vaccine acceptance among pregnant woman attending antenatal care clinic at Gondar town, Northwest Ethiopia, 2021 (n = 510).**

| Variables | COVID-19 Vaccine Acceptance | | COR (95% CI) | AOR (95% CI) |
|---|---|---|---|---|
| | Yes | No | | |
| **Maternal age (years)** | | | | |
| **18–24** | 41 | 69 | 1 | 1 |
| **25–34** | 144 | 225 | 1.08 (0.69, 1.67) | 0.72 (0.42, 1.23) |
| **≥35** | 26 | 5 | 8.75 (3.12, 24.57) | 5.68 (1.78, 18.17)** |
| **Place of residence** | | | | |
| **Urban** | 162 | 233 | 0.94 (0.62, 1.43) | |
| **Rural** | 49 | 66 | 1 | |
| **Marital status** | | | | |
| **Married** | 183 | 254 | 0.72 (0.31, 1.70) | |
| **Unmarried** | 17 | 32 | 0.50 (0.18, 1.39) | |
| Others[M] | 11 | 11 | 1 | |
| **Educational status** | | | | |
| **No formal education** | 42 | 67 | 1 | |
| **Primary education** | 68 | 91 | 1.19 (0.73, 1.96) | |
| **Secondary and above** | 101 | 141 | 1.14 (0.72, 1.82) | |
| **Occupation status** | | | | |
| **Employed** | 122 | 162 | 1.16 (0.81. 1.65) | |
| **Unemployed** | 89 | 137 | 1 | |
| **Gravidity** | | | | |
| **Primi-gravida** | 63 | 118 | 1 | 1 |
| **Multigravida** | 148 | 181 | 1.53 (1.05, 2.23) | 1.28 (0.80, 2.05) |
| **Number of ANC visit** | | | | |
| 1 time | 14 | 34 | 1 | |
| **2 times** | 66 | 108 | 1.48 (0.74, 2.97) | |
| **3 times** | 82 | 107 | 1.86 (0.94, 3.70) | |
| **4 times and above** | 49 | 50 | 2.38 (1.14, 4.97) | |
| **History of Abortion** | | | | |
| **Yes** | 42 | 29 | 2.31 (1.39, 3.86) | 1.43 (0.77, 2.66) |
| **No** | 169 | 270 | 1 | 1 |
| **Having a medical illness or pre-existing chronic disease** | | | | |
| **Yes** | 52 | 22 | 4.12 (2.41, 7.03) | 3.13 (1.70, 5.77)** |
| **No** | 159 | 277 | 1 | 1 |
| **Knowledge about COVID-19 vaccine** | | | | |
| **Poor knowledge** | 13 | 47 | 1 | 1 |
| **Good knowledge** | 198 | 252 | 2.84 (1.50, 5.40) | 2.39 (1.14, 5.00)* |
| **Attitude toward COVID-19 vaccine** | | | | |
| **Poor attitude** | 46 | 116 | 1 | |
| **Good attitude** | 165 | 183 | 2.27 (1.52, 3.40) | 2.13 (1.35, 3.36)* |
| **Having a history of contact with COVID-19 diagnosed people** | | | | |
| **Yes** | 34 | 11 | 5.03 (2.49,10.18) | 7.72 (2.18, 27.33)* |
| **No** | 177 | 288 | 1 | 1 |
| **The presence of family members or friends lost to COVID-19** | | | | |
| **Yes** | 28 | 13 | 3.37 (1.70, 6.67) | 0.63 (0.17, 2.35) |
| **No** | 183 | 286 | 1 | 1 |
| **Did have a test for COVID-19?** | | | | |

*(Continued)*

**Table 4.** (Continued)

| Variables | COVID-19 Vaccine Acceptance | | COR (95% CI) | AOR (95% CI) |
|---|---|---|---|---|
| | Yes | No | | |
| Yes | 79 | 87 | 1.46 (1.00, 2.12) | 0.91 (0.58, 1.42) |
| No | 132 | 212 | 1 | 1 |

**Note**: Others[M] includes (divorced, widowed and separated). 1 = Reference,

** = p< 0.001,

* = p< 0.05.

among the general population [28, 29, 43, 50–52]. However, as compared to studies conducted in Switzerland 29.7% [7], Southwest Ethiopia 31.3% [25], and 30.3% in Singapore [53], our finding is high. In addition to this, comparable findings were stated in Turkey [27], the United States (US) [54], and New York [55] where only 37%, 41%, and 44.3% of pregnant women were willing to accept the COVID-19 vaccine respectively. The inconsistency of these results could be explained by differences in socio-demographic characteristics and COVID-19 vaccine awareness levels among study participants, as well as study settings and time differences [51]. Furthermore, the discrepancy in the aforementioned results could be due to the fact that our study used face-to-face interviews, whilst the other studies used web-based surveys.

COVID-19 vaccine is nearly 5.7 times more likely to be accepted by pregnant women aged ≥35 years old as compared to pregnant women with ages between 18 and 24 years old. This finding was supported by studies in Switzerland [7], Ethiopia [23], Ecuador [56], and the US among the adult population [47]. This could be attributable to the fact that COVID-19 viral complications were more deadly in the elderly population than in the younger population. Furthermore, when people get older, they may develop age-related chronic conditions such as hypertension, kidney disease, and heart disease, which can lower a pregnant mother's immunity and raise the risk of COVID-19-related morbidity and death. As a result, it might create fear in the elderly population, leading to a greater willingness to accept the COVID-19 vaccine [23].

Good knowledge about the COVID-19 vaccine was also another main factor for COVID-19 vaccine acceptance. Pregnant women who had good knowledge about COVID-19 were nearly 2.4 times (AOR = 2.391, 95% CI: 1.144–4.998) more likely to accept the COVID-19 vaccine as compared to their counterparts. This could be explained by the fact that pregnant women with good COVID-19 vaccine knowledge were aware of the virus's severity to themselves and their fetuses so that they could simply accept the COVID-19 vaccine to diminish the pandemic's effects. Also, having good knowledge about the COVID-19 vaccine can help pregnant women in understanding the benefits of the COVID-19 vaccination program. This finding is supported by results from Ethiopia [23], New York [26], and Singapore [53] among pregnant women and by studies done in Ethiopia [29] and Egypt [57] among the general population.

The attitude of pregnant women toward the COVID-19 vaccine was the other most important predictor of COVID-19 vaccine acceptance. When compared to their counterparts, pregnant women with a positive attitude toward the COVID-19 vaccination were 2.1 times (AOR = 2.128, 95% CI: 1.348–3.360) more likely to be vaccinated against COVID-19. Similar findings were indicated among pregnant women [25, 58] and different study participants [28, 59] in Ethiopia and pregnant women in South Africa [60]. The reason for this could be that attitudinal factors such as worry about COVID-19 disease might increase the odds of accepting the vaccine [57, 61].

When compared to pregnant women without a pre-existing chronic disease, having a chronic disease would increase COVID-19 vaccine acceptability by 3.1 times, which is comparable with previous findings in Ethiopia [29], Egypt [57], Mozambique [62], and the US [63] among the general population. This is related to the nature of COVID-19 and concerns that people with chronic diseases. People with such health issues have been identified as being at the highest risk of acquiring COVID-19 and becoming seriously ill as a result of the infection, which can result in death. As a result, people with such health issues would work on all available preventative measures to avoid contracting COVID-19 infection [64].

Furthermore, similar to studies conducted among the general population in Mozambique [62] and Saudi Arabia [65], this study also found out pregnant women with a history of contact with COVID-19 diagnosed people were more likely to accept the vaccine (AOR = 7.724, 95% CI: 2.183–27.329) than those who had no history of contact with COVID-19 diagnosed people which is due to a concern of becoming (re)infected by COVID-19 [62]. However, no association was found between COVID-19 vaccine acceptance and participants who lost family members or friends when compared to those who did not lose any friends or family members. This finding necessitates further investigation. This study tried to assess the COVID-19 vaccine acceptance and associated factors among pregnant women attending antenatal care clinics. However, some limitations need highlighting. First: even though this study identified the factors associated with COVID-19 vaccine acceptance, we believe supplementing with a qualitative study could help to explore the socio-cultural barriers to pregnant mothers' refusal to accept the COVID-19 vaccine. Second, given it was a health facility-based study, the findings of the study may not be generalizable to the general population. Third, the nature of the study design, being cross-sectional, does not confer causal relationship. Fourth, although systematic random sampling takes less time than random sampling, it might lead to skewed results if the data set contains patterns. Hence, the results of our study need to be interpreted with caution.

## Conclusion

The findings of our study revealed that the acceptance of a COVID-19 vaccine was low. Acceptance of the COVID-19 vaccine was significantly associated with maternal age, having contact history with COVID-19 diagnosed people, knowledge and attitude about COVID-19 vaccine, and having a history of chronic illness. To enhance the COVID-19 vaccine acceptance, the government with different stakeholders should strengthen public education about the importance of getting COVID-19 vaccine.

## Supporting information

**S1 File. English version questionnaire.**
(PDF)

**S2 File. STROBE checklist.**
(DOC)

**S1 Data. Minimal data set used in the study.**
(SAV)

## Acknowledgments

Our heartfelt thanks go to the Health Department of Gondar town and each health facility for providing a permission letter. We would also like to acknowledge the study subjects, data collectors, and supervisors.

## Author Contributions

**Conceptualization:** Zewdu Bishaw Aynalem, Birhaneslasie Gebeyehu Yazew.

**Data curation:** Zewdu Bishaw Aynalem, Tewodros Worku Bogale, Getasew Mulat Bantie, Agumas Fentahun Ayalew, Workineh Tamir, Dejen Getaneh Feleke, Birhaneslasie Gebeyehu Yazew.

**Formal analysis:** Zewdu Bishaw Aynalem, Tewodros Worku Bogale, Getasew Mulat Bantie, Agumas Fentahun Ayalew, Workineh Tamir, Dejen Getaneh Feleke, Birhaneslasie Gebeyehu Yazew.

**Funding acquisition:** Zewdu Bishaw Aynalem, Birhaneslasie Gebeyehu Yazew.

**Investigation:** Zewdu Bishaw Aynalem, Tewodros Worku Bogale, Getasew Mulat Bantie, Agumas Fentahun Ayalew, Workineh Tamir, Dejen Getaneh Feleke, Birhaneslasie Gebeyehu Yazew.

**Methodology:** Zewdu Bishaw Aynalem, Tewodros Worku Bogale, Getasew Mulat Bantie, Agumas Fentahun Ayalew, Workineh Tamir, Dejen Getaneh Feleke, Birhaneslasie Gebeyehu Yazew.

**Project administration:** Zewdu Bishaw Aynalem, Birhaneslasie Gebeyehu Yazew.

**Resources:** Zewdu Bishaw Aynalem, Birhaneslasie Gebeyehu Yazew.

**Software:** Zewdu Bishaw Aynalem, Birhaneslasie Gebeyehu Yazew.

**Supervision:** Zewdu Bishaw Aynalem, Birhaneslasie Gebeyehu Yazew.

**Validation:** Zewdu Bishaw Aynalem, Tewodros Worku Bogale, Getasew Mulat Bantie, Agumas Fentahun Ayalew, Workineh Tamir, Dejen Getaneh Feleke, Birhaneslasie Gebeyehu Yazew.

**Visualization:** Zewdu Bishaw Aynalem, Birhaneslasie Gebeyehu Yazew.

**Writing – original draft:** Zewdu Bishaw Aynalem, Tewodros Worku Bogale, Getasew Mulat Bantie, Agumas Fentahun Ayalew, Workineh Tamir, Dejen Getaneh Feleke, Birhaneslasie Gebeyehu Yazew.

**Writing – review & editing:** Zewdu Bishaw Aynalem, Tewodros Worku Bogale, Getasew Mulat Bantie, Agumas Fentahun Ayalew, Workineh Tamir, Dejen Getaneh Feleke, Birhaneslasie Gebeyehu Yazew.

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
