## [Decision Letter · Decision Letter 0]

5 Jun 2022

PONE-D-22-06102Factors Associated With Willingness to Take COVID-19 Vaccine among Pregnant Women at Gondar Town, Northwest Ethiopia: A Multicenter Institution-Based Cross-Sectional StudyPLOS ONE

Dear Dr. Aynalem,

Thank you for submitting your manuscript to PLOS ONE. After careful consideration, we feel that it has merit but does not fully meet PLOS ONE’s publication criteria as it currently stands. Therefore, we invite you to submit a revised version of the manuscript that addresses the points raised during the review process. Editor's comments:

My understanding that “a lottery method” is a “random selection”. While “lottery method” might be understandable term, but not widely used. I would suggest using “random selection” instead.

Make sure the manuscript meets the PLOS One journal requirements in style and format.

It looks like the authors used a systematic random selection approach when recruiting participants. What are the limitations of such approach?

Who are the language experts? Bilingual translators?

More information is needed to understand reliability and validity of the questionnaire in the study population. Was the questionnaire pre-tested? If yes, how many participants responded? Any modifications were introduced? How experts assessed the validity of the items? What approach was undertaken to perform the expert review? Have the authors assess internal validity of the questionnaire? Have the authors performed confirmatory or exploratory factor analysis to assess interrelationships of items? Have the authors used any methods to validate the questionnaire? Have the authors accounted for the complex survey design in all statistical analyses? Without proper accounting for survey design, estimates and corresponding confidence intervals might be incorrectly calculated.

We look forward to receiving your revised manuscript.

Kind regards,

Alpamys Issanov

Academic Editor

PLOS ONE

Journal Requirements:

Reviewers' comments:

Reviewer's Responses to Questions

**Comments to the Author**

1. Is the manuscript technically sound, and do the data support the conclusions?

Reviewer #1: Yes

Reviewer #2: Yes

2. Has the statistical analysis been performed appropriately and rigorously? 

Reviewer #1: Yes

Reviewer #2: Yes

3. Have the authors made all data underlying the findings in their manuscript fully available?

Reviewer #1: Yes

Reviewer #2: Yes

4. Is the manuscript presented in an intelligible fashion and written in standard English?

Reviewer #1: Yes

Reviewer #2: Yes

5. Review Comments to the Author

Reviewer #1: I read the manuscript entitled "" by ??? et al with great interest and would like to congratulate the authors for their effort. However, there are some minor points to improve the quality of manuscript:

1) In introduction authors discuss previous reports of vaccination COVID among pregnant women as follow:

"The refusal to accept the COVID-19 vaccine is estimated to be high and greatly differs between

countries (19). For example, an online survey of pregnant women in 16 countries found that

COVID-19 vaccine acceptance was more than 80% in Mexico and India, but lower than 45% in

the United States, Australia, and Russia (20). In Switzerland (7), the United Kingdom (21), China

4 (22), and Ethiopia (23) the willingness to accept the COVID-19 vaccine among pregnant women

was found 62.1%, 29.7%, 77.4%, and 70.7% respectively (7, 21, 22). The main factors that

determine an individual’s intent to take the COVID-19 vaccine include gender, age, educational

status, occupational status, income, pre-existence of chronic disease, attitude towards and

knowledge of COVID-19 (23-28). Moreover, multiple myths and conspiracy theories on vaccines

and COVID-19 (5, 29, 30) would also potentially affect the COVID-19 vaccine acceptance."

Please add the recent meta-analysis published evaluating the global acceptance rate of covid-19 vaccine among pregnant women. PMID: 34670322 DOI: 10.1055/a-1674-6120

2) Page7 (methods) How about the history of TdP or flu vaccine in pregnancy and the chance of accepting COVID-19 vaccination?

3) Page 8 (methods) move ethics to the beginning of the methods.

4) Page 13 and 14. it is not clear for which variables the adjustment model has been performed?

Reviewer #2: Authors performed an interesting cross-sectional study in Northwest Ethiopia, exploring the acceptance rate of covid-19 vaccine among pregnant women. They observed a low rate of acceptance, and found that women would get it when there is enough knowledge and attitude towards the vaccine. Indeed, this is the proof that when people tend to get information, the fear of unknown and unproven consequences lowers, with the effect of favoring higher uptake of vaccines. Therefore, governments should push on informative politics and campaigns with reinforcement of positive news about vaccines against SARS-CoV-2 as recent literature is showing.

The study is quite well conducted and reported. However, here are some remarks:

- there are no line numbers and this makes difficult to refer to certain sentence. When authors will resubmit the manuscript, please use line numbers

- please reformulate the methods section in the abstract which although quite understandable is not very clearly written and difficult to read especially in relation to statistical analysis, as well as the corresponding section of methods in the manuscript

- The discussion needs to be implemented: first of all, authors referred to various surveys conducted through the world, but missed to mention the first two performed in Italy, which was one of the first country that faced the diffusion of the infection after China (Carbone L, Mappa I, Sirico A, Di Girolamo R, Saccone G, Di Mascio D, Donadono V, Cuomo L, Gabrielli O, Migliorini S, Luviso M, D’antonio F, Rizzo G, Maruotti GM. Pregnant women perspectives on SARS-COV-2 vaccine. Am J Obstet Gynecol MFM. 2021 Mar 23:100352. Doi: 10.1016/j.ajogmf.2021.100352. Epub ahead of print. ----- Mappa I, Luviso M, Distefano FA, Carbone L, Maruotti GM, Rizzo G. Women perception of SARS-CoV-2 vaccination during pregnancy and subsequent maternal anxiety: a prospective observational study. J Matern Fetal Neonatal Med. 2021 Apr 11:1-4. doi: 10.1080/14767058.2021.1910672. Epub ahead of print.)

- In addition, a systematic review found more or less a similar percentage of uptake, and well discuss mechanism for the implementation of vaccine uptake (Shamshirsaz AA, Hessami K, Morain S, Afshar Y, Nassr AA, Arian SE, Asl NM, Aagaard K. Intention to Receive COVID-19 Vaccine during Pregnancy: A Systematic Review and Meta-analysis. Am J Perinatol. 2021 Oct 20. doi: 10.1055/a-1674-6120. Epub ahead of print. PMID: 34670322.)

- in this regard, in the last part of the discussion, authors could focus on how local policies could be implemented according to local healthcare issues and pregnancy management facilities

Corrections:

- introduction page 3, “in the movement”: please reformulate

- Introduction page 3, “cesarian” should be “cesarean”

- introduction page 3, “such the above-mentioned” could be “the abovementioned”

- introduction page 3, “particularly in pregnancy continues….” Please reformulate

- please delete the ORCID URL placed in page 5, in the middle of the methods, before figure 1

- methods, Data collection instrument and Study variables, page 7, “by reviewing different literature” could more properly be “after exhaustive literature search”

- methods, Data collection instrument and Study variables, page 7, “it was the outcome variable of the study” should be “it represents the main outcome of the study”

- discussion, page 15, I suggest to not use the term “perfect solution” when referring to the vaccine against SARS-CoV-2. Indeed, is a very useful and protective tool which has demonstrated to be effective in reducing the risk of contracting the infection but also reducing the severity of the eventual illness. However, “perfect solution” seems too much

- please report the references consistently among them and as per journal guidelines

6. PLOS authors have the option to publish the peer review history of their article (what does this mean?). If published, this will include your full peer review and any attached files.

Reviewer #1: **Yes: **Kamran Hessami

Reviewer #2: No

---

## [Author Response · Author response to Decision Letter 0]

21 Jun 2022

Reviewer 1: Thank you for your time and effort! All comments and recommendations are helpful and beneficial to the paper's progress. All the comments have been addressed.

Reviewer 2: Thank you for your time and effort! All comments and suggestions are constructive and important for the improvement of the paper. All comments are addressed.

---

## [Editor Report · Decision Letter 1]

9 Aug 2022

PONE-D-22-06102R1Factors Associated With Willingness to Take COVID-19 Vaccine among Pregnant Women at Gondar Town, Northwest Ethiopia: A Multicenter Institution-Based Cross-Sectional StudyPLOS ONE

Dear Dr. Aynalem,

Thank you for submitting your manuscript to PLOS ONE. After careful consideration, we feel that it has merit but does not fully meet PLOS ONE’s publication criteria as it currently stands. Therefore, we invite you to submit a revised version of the manuscript that addresses the points raised during the review process.

 Editor's comments:I would recommend expanding on this statement “Third, the nature of the study design, being cross-sectional, does not show the exact connection between cause and effect.” Why do the authors believe that it is challenging to determine causality?Please add limitations of the systematic random sampling in the limitations section.Numbers in the tables can be rounded to one or two decimal places.What was the conceptual framework for examining the research question? I do see that the independent variables were selected based on the threshold of p<0.2. But have the authors considered epidemiologically, conceptually important variables to be included in the model? Or was the analysis purely explorative/descriptive? I would encourage to include a visual representation of the theoretical framework of the study so a reader can relate to it.When the authors stated that “[variables]…were entered into the multivariable analysis to control for possible confounders”, it seems a bit counterintuitive to the research question. The study aims to find factors associated with the vaccine acceptance, while adjusting for confounding factors usually is done in the causative research question (one primary exposure variable). I would recommend aligning the research question with the selected analysis.I also noticed that confidence intervals for some independent variables were extremely large. For example, for age 35+. Have the authors attempted to perform sensitivity analysis with different categorization of age?While I appreciate the authors adding information on how the survey tool was developed, I am still concerned about the validity and reliability of the tool. For example, the authors stated that the reliability of the tool was tested using the Cronbach’s alpha which was equal to 0.87. However, my understanding is that there are several constructs in the questionnaire and one reliability measurement is not enough.I am also curious how the final 10 questions for vaccine knowledge and attitude were derived? What were the initial questions and what were the number of questions? How were they selected?Have the authors considered different thresholds when selecting for attitude and vaccine knowledge constructs? What were the results?In the analytical part, have the authors considered potential multicollinearity in the analysis? Some of the constructs may have overlapping areas that could cause collinearity. Whether this concern could have affected the results?Please include the questionnaire as a supporting document.

We look forward to receiving your revised manuscript.

Kind regards,

Alpamys Issanov

Academic Editor

PLOS ONE
---

## [Author Response · Author response to Decision Letter 1]

23 Sep 2022

All comments and suggestions are very important and constructive. All comments are addressed. Thank you!

---

## [Editor Report · Decision Letter 2]

13 Oct 2022

Factors Associated With Willingness to Take COVID-19 Vaccine among Pregnant Women at Gondar Town, Northwest Ethiopia: A Multicenter Institution-Based Cross-Sectional Study

PONE-D-22-06102R2

Dear Dr. Aynalem,

We’re pleased to inform you that your manuscript has been judged scientifically suitable for publication and will be formally accepted for publication once it meets all outstanding technical requirements.

Kind regards,

Alpamys Issanov

Academic Editor

PLOS ONE
---

## [Editor Report · Acceptance letter]

20 Oct 2022

PONE-D-22-06102R2 

Factors Associated With Willingness to Take COVID-19 Vaccine among Pregnant Women at Gondar Town, Northwest Ethiopia: A Multicenter Institution-Based Cross-Sectional Study 

Dear Dr. Aynalem:

I'm pleased to inform you that your manuscript has been deemed suitable for publication in PLOS ONE. Congratulations! Your manuscript is now with our production department. 

Kind regards, 

on behalf of

Dr. Alpamys Issanov 

Academic Editor

PLOS ONE